# Predicting Disease in Transition Dairy Cattle Based on Behaviors Measured Before Calving

**DOI:** 10.3390/ani10060928

**Published:** 2020-05-27

**Authors:** Mohammad W. Sahar, Annabelle Beaver, Marina A. G. von Keyserlingk, Daniel M. Weary

**Affiliations:** 1Animal Welfare Program, Faculty of Land and Food Systems, University of British Columbia, Vancouver BC V6T 1Z4, Canada; m.wali_sahar@yahoo.com (M.W.S.); abeaver@harper-adams.ac.uk (A.B.); nina@mail.ubc.ca (M.A.G.v.K.); 2Department of Animal Production, Welfare and Veterinary Sciences, Harper Adams University, Shropshire TF10 8NB, UK

**Keywords:** animal welfare, aggressive interaction, eating behavior, transition period

## Abstract

**Simple Summary:**

Dairy cattle often become ill after calving. We developed models designed to predict which cows are likely to become ill based upon measures of the cows’ feeding and competitive behaviors before calving. Our models had high sensitivity (73–71%), specificity (80–84%), positive predictive values (73–77%), and negative predictive values (80–80%) for both cows that had previously calved and for those calving for the first time. We conclude that behaviors at the feed bunk before calving can predict cows at risk of becoming sick in the weeks after calving.

**Abstract:**

Dairy cattle are particularly susceptible to metritis, hyperketonemia (HYK), and mastitis in the weeks after calving. These high-prevalence transition diseases adversely affect animal welfare, milk production, and profitability. Our aim was to use prepartum behavior to predict which cows have an increased risk of developing these conditions after calving. The behavior of 213 multiparous and 105 primiparous Holsteins was recorded for approximately three weeks before calving by an electronic feeding system. Cows were also monitored for signs of metritis, HYK, and mastitis in the weeks after calving. The data were split using a stratified random method: we used 70% of our data (hereafter referred to as the “training” dataset) to develop the model and the remaining 30% of data (i.e., the “test” dataset) to assess the model’s predictive ability. Separate models were developed for primiparous and multiparous animals. The area under the receiver operating characteristic (ROC) curve using the test dataset for multiparous cows was 0.83, sensitivity and specificity were 73% and 80%, positive predictive value (PPV) was 73%, and negative predictive value (NPV) was 80%. The area under the ROC curve using the test dataset for primiparous cows was 0.86, sensitivity and specificity were 71% and 84%, PPV was 77%, and NPV was 80%. We conclude that prepartum behavior can be used to predict cows at risk of metritis, HYK, and mastitis after calving.

## 1. Introduction

During the transition period, defined as ± three weeks around calving [1], dairy cows experience a number of physiological and environmental challenges [2], and many succumb to ketosis, metritis, and mastitis [1,3,4]. The percentage of cows affected by ketosis or subclinical ketosis (SCK) ranges from 11% to 49% during the first two months of lactation [5,6]. The onset of ketosis is associated with a number of factors, including rapid fetal growth in late pregnancy, high milk yield in early lactation, and increasing energy demand [7]. If feed intake is insufficient to meet demand, cows will mobilize body fat, increasing the risk of ketosis [8,9,10]. Metritis (uterine inflammation caused by bacterial infection) affects between 21% and 40% of dairy cattle during the first weeks after calving [11,12,13]. It generally occurs shortly after calving, with a higher risk associated with twin birth, stillbirth, dystocia, retained placenta, and reduced body condition score (BCS) [10,14,15]. Mastitis is caused predominately by bacterial infection of the udder [16]; the incidence rate of clinical mastitis in Canada has been estimated to be 23/100 cow years [17].

Each of these conditions is associated with reduced reproductive performance and milk production, decreased milk quality, and an increased risk of other diseases and culling [18,19]. Metritis and mastitis are also known to cause pain [20,21]. Developing tools for the early identification of animals at risk of transition-period diseases may help in prevention and treatment.

Feeding and agonistic behavior can be useful in identifying animals at risk of postpartum diseases. Reduced feed intake and time spent feeding before calving are associated with metritis [22] and ketosis [23] after calving. Reduced feed intake is also associated with mastitis [24]. Decreased agonistic behavior prepartum is associated with increased disease risk postpartum for metritis [22] and ketosis [23]. Previous research has also shown that ketosis is more likely in over-conditioned cows [25], which may relate to feeding behavior. 

Previous research [24,26,27] has been designed to identify variables associated with diseases. The current study builds upon these previous works by developing predictive models. Our aim was to evaluate the odds of dairy cattle developing one or more of three common transition diseases (metritis, hyperketonemia, or mastitis) during the postpartum period based upon prepartum indicators. In addition, previous studies have typically used a single category of predictor variables; for instance, one study [28] evaluated the relationship between lying behavior and transition cow disease, and another study [26] investigated the relationship between prepartum feeding time and postpartum disease. In contrast, the current study uses multiple categories, including prepartum feeding behavior, agonistic behavior, and feed intake, to predict postpartum diseases. 

## 2. Materials and Methods 

The study was conducted at the University of British Columbia’s (UBC) Dairy Education and Research Center in Agassiz, BC. All animal-related procedures were conducted in compliance with approved animal-care protocols from UBC (A15-0084). This study used the same animals reported in previous articles [29,30,31,32]. After diagnosis with metritis, the animals were treated with either meloxicam (0.5 mg/kg; Metacam 20-mg/mL solution; Boehringer Ingelheim GmbH, Ingelheim am Rhein, Germany) or a placebo solution (Boehringer Ingelheim GmbH). Our study was solely concerned with prepartum predictors of disease, and thus, it was not necessary to account for these treatments in our models. 

We enrolled 337 Holstein dairy cows 3 weeks before the expected calving date, of which 232 were multiparous and 105 primiparous (average ± SD parity of 1.8 ± 1.9; range of 0 to 8 lactations). Cows with a gait score of >3 (scored following [33]) in the week before enrolment were not included. Nineteen multiparous cows were later excluded due to missing data. 

Fresh and dry cows were housed in separate pens, each with 24 freestalls at a stocking density that did not exceed 20 cows/pen at any given time. The dimensions of the stalls were as follows: 2.6 × 1.2 m, neck rail 1.2 m above stall surface, separated by freestall partitions (Y2K partitions; Artex, Langley, BC, Canada). Each stall had 5 cm of sand bedding on top of a mattress base (Pasture Mat; Promat Inc., Woodstock, ON, Canada). Each pen was equipped with 12 Insentec feed bins (Insentec, Marknesse, The Netherlands) and 2 Insentec water troughs. Three weeks before the expected calving date, cows were brought to the prepartum pen and, after calving signs were observed, were moved to the maternity pen. The maternity pen was equipped with 6 Insentec feed bins and a water trough, with a sawdust-bedded pack for a lying surface. The number of cows in the maternity pen never exceeded 2 at a given time. Cows were moved to the postpartum pen 24 h after calving and monitored for the next 3 weeks. Cows were milked in a double 12 parallel milking parlor twice per day at approximately 07:00 and 17:00 h, with waiting time in the holding area never exceeding 30 min. Cows were fed a total mixed ration (TMR) twice a day at approximately 08:00 and 16:00 h (see [32] for additional information). Feed was formulated [34] to meet or exceed nutrient requirements of a Holstein cow producing 40 kg/d of 3.5% fat-corrected milk (FCM) and weighing 620 kg. 

### 2.1. Health Assessment

Cows were observed for signs of disease for 21 d after calving. Cows were checked for metritis on days 3, 6, 9, 12, and 15 after calving. Vaginal discharge was scored based on appearance and smell on a scale of 0 to 4, as described by [22], where 0 = clear or no mucus; 1 = cloudy mucus, red or brown mucus, or mucus with flecks of pus and no foul smell; 2 = mucopurulent (≤50% pus present) and foul-smelling; 3 = purulent (≥50% pus present) and foul-smelling; and 4 = putrid (red or brown color, watery, foul-smelling). Cows were considered metritic if their vaginal discharge scored 2, 3, or 4 on at least one occasion. After diagnosis, cows were treated with ceftiofur (2.2 mg/kg of body weight (BW) (subcutaneous); Excenel RTU sterile suspension, 50 mg/mL as ceftiofur hydrochloride; Zoetis, Parsippany, NJ, USA) injected once per day after morning milking for 5 consecutive days and checked for metritis every 3 d until 21 d after calving. 

Hyperketonemia (HYK) was diagnosed by farm staff or the herd veterinarian using a milk beta-hydroxybutyric (BHB) acid test strip result ≥100 μmol/L (KetoTest; Elanco Animal Health, Nagoya, Japan) when feed intake was low or milk production was reduced. Cows were checked for clinical mastitis by farm staff twice per day based on udder characteristics (red, swollen, and/or hard) and clots in milk. Cows that did not get sick with HYK, metritis, or mastitis during this period were recorded as healthy for analysis. There were 15 lame cows (scoring >3 following [33]). Nine of these did not have HYK, metritis, or mastitis; these animals were coded as healthy for analysis.

### 2.2. Behavioral Data Collection

Feeding behaviors (time spent feeding, feed intake, and number of visits) and agonistic behaviors were automatically recorded by the Insentec electronic feed monitoring system. A visit was recorded each time the cow arrived at, and left, the feed bunk. Measurements such as the number of meals and number of visits per meal were calculated from the data derived from the Insentec system using the criterion developed previously [35]. Agonistic interactions at the feed bunk were detected automatically using the algorithm (see [36]). Water intake data and agonistic behaviors at the water bins were collected using the Insentec electronic water monitoring system [37]. 

### 2.3. Statistical Analysis

Statistical analysis was performed using SAS (version 9.4; SAS Institute Inc., Cary, NC, USA). ROC curve figures were generated using SAS and Kolmogorov-Smirnov (KS) figures using Excel (version 12.0). 

#### 2.3.1. Data Preparation

Multiple observations recorded for each cow were averaged for each variable. Data were split into datasets containing 70% and 30% of observations, according to a stratified random sampling procedure [38]. The stratification was conducted as follows: cows were first grouped as (1) healthy (cows that did not get sick during the first 17 days postpartum); (2) having developed 1 condition (either metritis, HYK, or mastitis) during the first 17 d postpartum; or (3) having developed ≥2 conditions during the first 17 d postpartum. Each subset of data was then split at the ratio of 70:30. Datasets containing 70% of the observations were then merged together. The same procedure was applied to the datasets with 30% of the observations. Each dataset was then divided into primiparous and multiparous animals, resulting in 4 unique datasets. Following this split, the amount of data contained in each of the 4 datasets remained approximately 70% or 30%. Multiparous and primiparous cows were separated due to differences in feed intake, body weight, bins visited per meal, and time spent feeding. For each parity category, the 70% subset was employed for model development (“training”), and the 30% subset was used to test the model’s predictive ability. Data collected from the water bins were also split 70:30 according to the stratified random sampling procedure described previously. 

#### 2.3.2. Descriptive Analysis

The training datasets were used for all model development stages, including the initial descriptive analysis. All variables with the potential to predict postpartum disease were considered for inclusion. These variables were feed intake (DMI), time spent eating, number of meals per day, bins visited per day, lameness, BW, and the number of agonistic interactions during the prepartum period. Agonistic interactions were defined as any aggressive interactions between individuals, characterized by “actor behaviors” (in which the subject replaces another cow at the feed bunk) and “reactor behaviors” (in which the subject is displaced and replaced by another cow at the feed bunk). The distributions of continuous variables were explored using PROC Univariate, and correlations between these variables were assessed using PROC Corr. Variables were considered correlated at a Pearson’s r value of ≥0.7, and this information was later used during the model building phase to avoid multicollinearity. 

#### 2.3.3. Model Development

To predict the prevalence of disease, multivariable binary logistic regression models were developed using the predictors of interest and postpartum disease status as the outcome. Separate models were developed and tested for both primi- and multiparous cows. Animals were classified as sick if they were diagnosed postpartum with at least one of the following conditions: metritis, HYK, and mastitis. Due to the limited number of variables, predictors were not screened at the univariable level. Univariable models were only used to select among pairs of correlated variables; the variable most related to the outcome was selected for inclusion in the multivariable models. Variables were removed from the multivariable models using a manual backwards stepwise process where the least significant variable was sequentially removed. All relevant two-way interactions were considered. The final models contained variables meeting the *p* < 0.05 threshold. In cases where a variable did not meet the threshold of *p* < 0.05 but removing it from the model reduced the area under the ROC curve by 3% or more, we considered the Akaike information criterion (AIC) and Schwarz criterion (SC) values and conducted a nested −2 log likelihood test with and without the variable of interest. Further, we conducted ROC contrast tests to assess whether inclusion of that variable statistically improved the area under the ROC curve. We removed the variable from the model when none of these tests was significant. Hosmer and Lemeshow Goodness of Fit test (applied with the “lackfit” statement) was used to assess the model fit. Once the final models were developed using the training dataset, the same model was applied to the test datasets to assess the predictive ability. Water data were analyzed employing the same procedure; however, due to the weak predictive ability of the resulting model, results are reported only in the Appendix A (see Appendix A).

Odds ratios for feed intake, time spent feeding, and actor behaviors were calculated based on a 1-kg increase in feed intake, a 15 min increase in time spent feeding, and 6 additional actor behaviors, respectively [22,39]. Odds ratios for prepartum BW, number of bins visited per meal, and number of different bins visited per day were calculated based on 25 kg of BW, 1 additional bin visited per meal, and 1 new bin visited per day, respectively. 

We explored interactions graphically. To investigate the interaction between feed intake and time spent feeding for primiparous cows, we conducted additional analyses on the cows in the 25th and 75th percentiles for feed intake. Using PROC logistic, we compared sick and healthy cows within these subsets based on variables such as time spent feeding, prepartum BW, agonistic interactions, and bins visited per meal using univariable models.

To better understand the models’ ability to differentiate between sick and healthy animals, using the test datasets, we extracted the exact probabilities of becoming sick for each animal, as predicted by the models (using the OUT statement in SAS), and ranked these probabilities from highest to lowest to compare the distributions of healthy and sick animals via the Kolmogorov Smirnov (KS) statistic. The datasets and analysis code are provided in the Appendix A.

## 3. Results

### 3.1. Descriptive Statistics

The training dataset for multiparous cows contained 143 animals with mean parity of 2.5 ± 1.8 (SD), including 62 sick animals. Of these, 36 were metritic, 20 had HYK, 16 were mastitic, and 10 animals had a combination of two conditions. The test dataset for multiparous cows contained 70 animals with mean parity of 2.7 ± 1.8 (SD), including 17 cows with metritis, 11 with HYK, and 8 with mastitis; 11 cows had a combination of two conditions. 

The training dataset of primiparous cows contained 72 cows—of which 36 were metritic, 3 had HYK, and 3 were mastitic. Two cows had a combination of diseases. The testing dataset of primiparous cows contained 33 cows—of which 12 were metritic, 3 had HYK, and none were mastitic. One cow had two of the conditions. Table 1 shows the descriptive statistics for variables that were retained in the models developed for multi- and primiparous cows. 

### 3.2. Model Results

There was no lack of fit in the models developed for multiparous (*p* = 0.53) or primiparous (*p* = 0.42) cows.

For multiparous cows, higher prepartum BW and involvement in additional actor behaviors increased the odds of postpartum disease (Table 2). We found interactions between DMI and bins visited/day and between bins visited/day and bins visited/meal. In reference to the first interaction, when bins visited/day was low, a high feed intake improved the odds of remaining healthy; however, when the bins visited/day was high, feed intake did not influence the odds of disease. The second interaction was a crossover: increased bin visits/day was protective against becoming ill but only if the number of bins/meal was high.

For primiparous cows, the odds of becoming sick postpartum increased as the number of prepartum bin visits per meal increased. We found an interaction between time spent feeding and feed intake. In this case, increased time spent feeding was protective against becoming sick but only when feed intake was low (see Figure 1A). In contrast, the probability of becoming sick increased as time spent feeding increased in cows that had high feed intakes (see Figure 1B). To better understand why this may be happening, we evaluated the 25% cows with the highest feed intakes. Cows that were sick spent more time feeding (*p* = 0.04) and were replaced numerically more often from the feed bunk by other cows (*p* = 0.09) compared to those that remained healthy. Cows that became sick also visited more bins per meal (*p* = 0.05). We also evaluated the quartile of cows that spent the least amount of time feeding; for these cows, the only predictor of sickness was a reduced feed intake (*p* = 0.03).

#### Predictive Ability of the Models 

The logistic regression models predicted the development of postpartum disease for both multi- and primiparous cows. The areas under the ROC curves for the training and test datasets in multiparous cows were 83.5% and 83.4%, respectively (Figure 2A,B). The sensitivity and specificity of the model in the training dataset were 66.1% and 81.3%, respectively; the positive predictive value (PPV) was 73.2%, and the negative predictive value (NPV) was 75.6%. For the test dataset, sensitivity and specificity were 73.3% and 80.0%, respectively, and PPV and NPV were 73.3% and 80.0%, respectively. The overall accuracy of the model in the training and test datasets were 74.8% and 77.1%, respectively.

The areas under the ROC curves for the training and test datasets in primiparous cows were 75.2% and 85.7%, respectively (Figure 2C,D). The sensitivity and specificity of the model in the training dataset were 75.0% and 56.3%, respectively. The PPV of the model using the training dataset was 68.2%, and NPV was 64.3%. For the test dataset, sensitivity and specificity were 71.4% and 84.2%, and the PPV and NPV were 76.9% and 80.0%. The overall accuracy of the model in the training and the test datasets were 67.1% and 78.8%, respectively. 

The ability of the model to separate healthy and sick multiparous cows in the test dataset is illustrated using KS statistics (Figure 3A). For example, the first decile (representing the top 10% of animals at risk of becoming ill) includes approximately 16.7% of sick animals and 5.0% of healthy animals. The maximum separation point between the cumulative percentage of sick and healthy animals is found at the 6th decile; this maximum separation point thus accounts for 60% of all animals (including 96.7% of sick animals). Similarly, Figure 3B visually shows the ability of the model to separate healthy and sick primiparous cows in the test dataset using KS statistics. The maximum separation point between the cumulative percentage of sick and healthy animals is found at the 3rd decile; this maximum separation point thus accounts for 30% of all animals (including 64.3% of sick animals).

## 4. Discussion

Previous research has examined the association between feeding behavior and transition-period diseases. For example, reduced feed intake is associated with mastitis, ketosis, and metritis [24,40,41]. Similarly, eating time is associated with mastitis [42], ketosis [43], and metritis [22,44]. Previous work has also shown that cows that engaged in fewer agonistic interactions during the prepartum period were more likely to develop metritis [22] and ketosis [23] during the postpartum period. Moreover, cows that developed metritis postpartum were displaced from the feed bunk more often and displaced other cows less frequently during the prepartum period [27]. Cows later diagnosed with mastitis also engaged in fewer replacements from the feed bunk during the five days before diagnosis [24]. 

In contrast to the associative focus of previous studies, the current study specifically examined the predictive power of these measures in identifying animals likely to succumb to these ailments. Our results show that animals at risk of becoming sick postpartum can be identified with a reasonable degree of success (area under the ROC curve: 0.83–0.86). Multiparous cows that became sick postpartum were involved in a higher number of agonistic interactions. Cows that ate more feed and visited more bins per day had greater odds of remaining healthy postpartum, although an interaction revealed that a higher feed consumption (compared to increased visits to different bins visits per day) made a more substantive contribution to the odds of remaining healthy. Moreover, an increased number of bins visited per meal was only protective against postpartum disease if the number of different bins visited/day was low. Primiparous cows that became sick visited more bins per meal, potentially because they were replaced more often. The interaction between time spent feeding and feed intake revealed that, when feed intake was low, increased time spent feeding improved the odds of an animal remaining healthy postpartum. The only result in our study that disagreed with previous findings was the higher number of agonistic interactions in multiparous cows that became sick; earlier work has shown the opposite [22,23,24]. We suggest that the relationship in the current study was influenced by BW, as cows that became sick had higher BW compared to those that remained healthy, and increased BW is associated with social dominance in cows [45].

The use of predictive analysis is common in human medical literature. A systematic review article [46] reported results from 18 papers (using train-test splits or bootstrapping techniques)—of which 11 used predictive logistic regression to detect prediabetic patients. Areas under the ROC curves in these models ranged from 69–75%, and the reported sensitivity and specificity ranged from 72.1–78% and 52–57%, respectively. Another systematic review [47] examined 71 papers (of which 41% divided their data into train-test or train-validation sets, 35% used cross-validation, 1.4% used bootstrapping, 12.6% used random splitting, and 10% used some form of external validation) to compare logistic regression with machine learning for clinical prediction models; the area under the ROC curve ranged from 52% to 99%, and the study found no evidence that algorithms based on machine learning resulted in improved predictions compared to logistic regression. 

The use of predictive analysis is also gaining traction in dairy science. A recently published study [48] used logistic regression applying a 10-fold cross-validation technique to detect lameness, resulting in an area under the ROC curve of 94%. Another study [49] fitted logistic regression models to predict lameness incidences in subsequent lactation using the data collected at the cessation of lactation. The area covered by the ROC curve was 77%, and overall accuracy of the model was reported as 74%. Compared to the models reported in both human health and dairy science sectors, our models achieved similar predictive success, with ROC curve values ranging from approximately 83–86%, PPV of 73–76%, and NPV of 80–80% (for multiparous and primiparous animals, respectively).

Preventing diseases in dairy cattle is important, as metritis [21] and mastitis [50] are painful conditions, and transition diseases impose substantial costs upon the farm. The cost per case of hyperketonemia (HYK) is estimated to be $117 USD [51]. The cost per case of mastitis is estimated to reach €198 [52], and the cost per case of metritis is estimated to be $106 USD [53]. According to our model, the 60% of multiparous cows with the highest probability of becoming ill postpartum (based upon prepartum indicators) will include approximately 97% of the cows that will actually become sick. The ability to predict which cows are at the highest risk of becoming sick offers farmers an opportunity to prevent these sicknesses from occurring. 

A practical limitation to applying these models is the difficulty that producers face in obtaining behavioral measurements. Automation of data collection of the variables considered here could be achieved by installing automatic feed bins, but these may not be practical for many farmers. More economical alternatives could include the use of sensors to monitor rumination or water intake (as fewer drinking spaces are required per cow). Unfortunately, the model developed from the water data had a poor ability to detect cows at high risk of becoming sick during the postpartum period, as shown in the Supplementary Information. 

The KS analysis provides a statistical cutoff that may be useful in deciding which animals to treat, but the biological significance of this threshold needs to be validated. Moreover, our data address three major transition diseases, but the model does not determine which of these diseases each at-risk animal is likely to get; more data are required in order to disentangle these diseases. The practical implications of our model must also be examined prior to commercial applicability. For example, the KS graph for primiparous animals shows that, at the maximum separation point, 64% of sick cows are accurately detected, and 5% of healthy cows are misclassified as sick. This maximum separation point may be statistically meaningful, but cutoff points at other deciles could prove to be more cost-effective or practical to implement. 

## 5. Conclusions

We developed models to predict postpartum disease in multiparous and primiparous dairy cows based upon measures of behavior before calving. Multiparous cows with increased BW and more actor behaviors were more likely to succumb to metritis, HYK, and mastitis postpartum. We also uncovered interactions between feed intake, different bins visited/day, and the number of bins visited/meal. For primiparous cows, key predictors of disease were more bins visited per meal and an interaction between feed intake and time spent feeding during the prepartum period, where increased time spent feeding was protective against becoming sick when the feed intake was low, and the probability of becoming sick increased as time spent feeding increased when feed intake was high. Predictive models successfully distinguished the majority of the sick animals from healthy animals. We conclude that prepartum behaviors can be used to predict animals at risk of transition cow diseases, improving our capacity for prevention and treatment.

## Figures and Tables

**Figure 1 animals-10-00928-f001:**
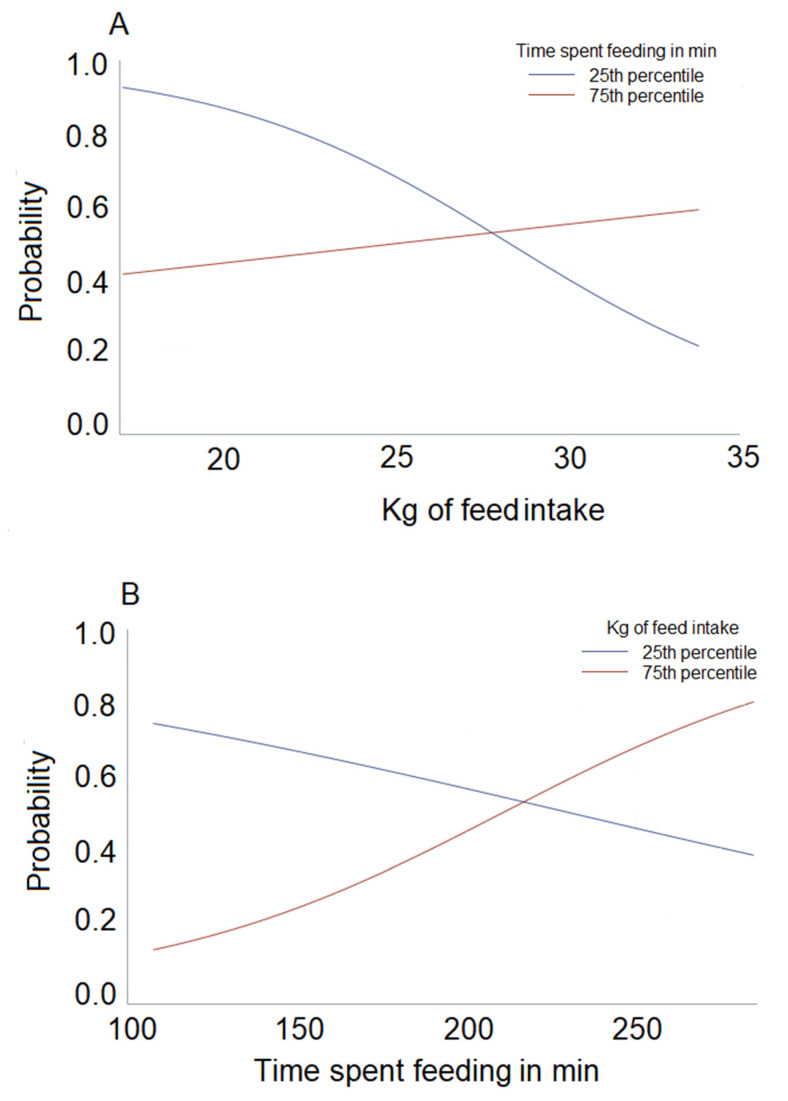
Panel (**A**) shows the predicted probabilities for postpartum disease in primiparous animals on the Y-axis and feed intake on the X-axis. The solid blue line represents a primiparous cow that spent 185 min feeding (i.e., the 25th percentile), and the red line represents a primiparous cow that spent 222 min of feeding (i.e., the 75th percentile). Panel (**B**) shows the predicted probabilities for postpartum disease in primiparous animals on the Y-axis and time spent feeding on the X-axis. The solid blue line represents a primiparous cow that consumed 26.6 kg of feed (i.e., the 25th percentile), and the red line represents a primiparous cow that consumed 30.3 kg (i.e., the 75th percentile). For both panels, the variable bins per meal was held constant at the mean value of 5.02.

**Figure 2 animals-10-00928-f002:**
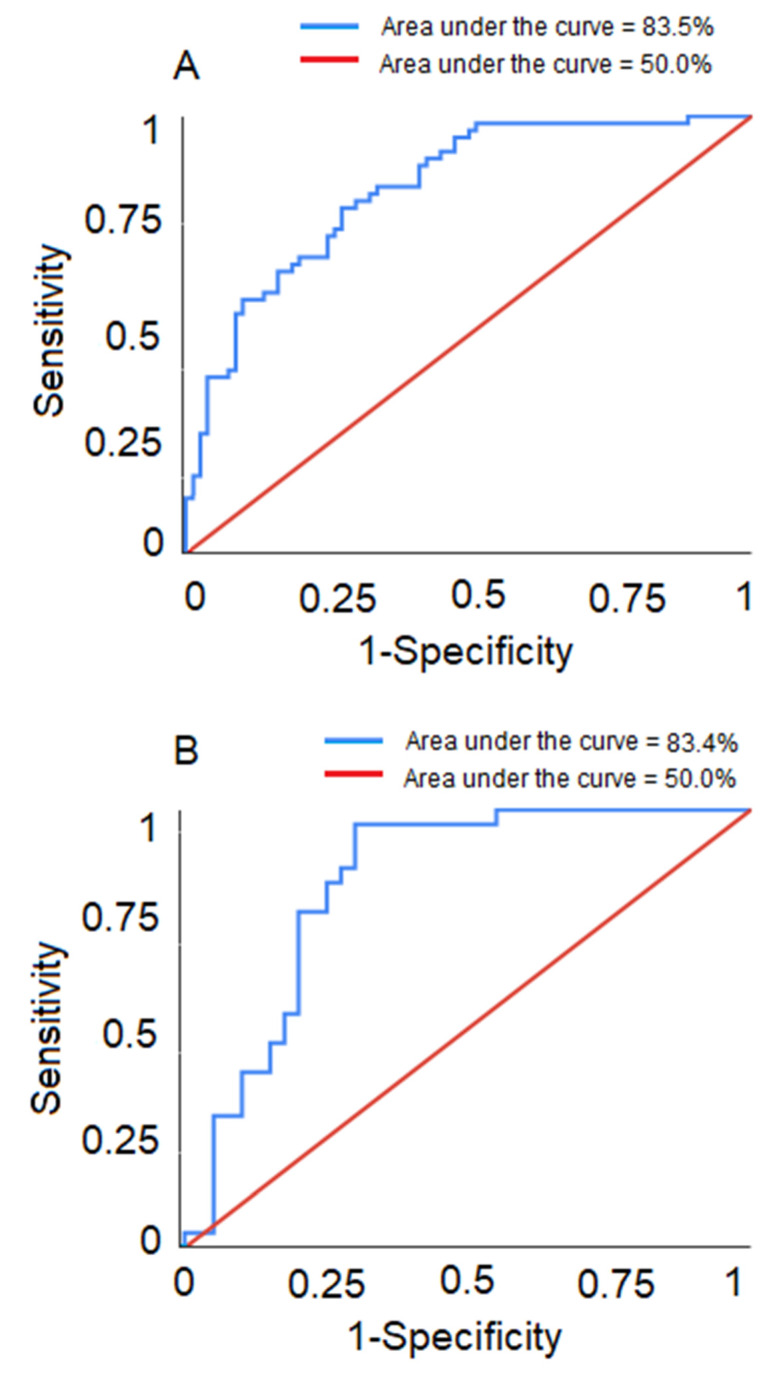
Receiver operating characteristic (ROC) curves for the logistic regression models shown separately for (**A**) the training dataset for multiparous cows, (**B**) the test dataset for multiparous cows, (**C**) the training dataset for primiparous cows, and (**D**) the test dataset for primiparous cows.

**Figure 3 animals-10-00928-f003:**
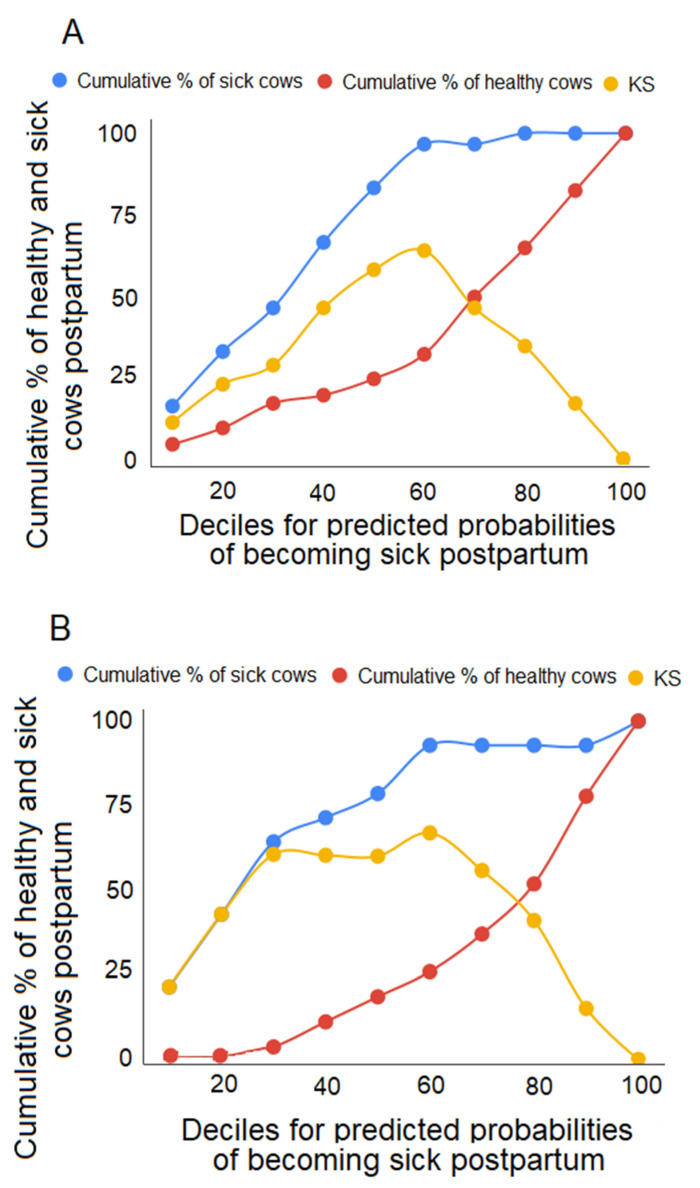
These Kolmogorov Smirnov (KS) graphs for multiparous (**A**) and primiparous (**B**) cows, calculated based upon the individual predicted probabilities of the logistic regression model. The blue lines and red lines indicate the cumulative percentage of sick ^1^ and healthy ^2^ cows in each decile, and the yellow lines represent the associated KS statistic (calculated as the difference between the cumulative percentages). ^1^ Cows that became sick with at least one disease (ketosis, metritis, or mastitis) during the postpartum period. ^2^ Cows that did not develop ketosis, metritis, or mastitis during the postpartum period.

**Table 1 animals-10-00928-t001:** Mean and SD of significant variables, shown separately for multiparous and primiparous cows, and by health status postpartum (i.e., healthy ^1^ or sick ^2^ with metritis, hyperketonemia (HYK), or mastitis). The data are for the combined training ^3^ and testing ^4^ datasets

Variable Name	Healthy		Sick	
	Mean	SD	Mean	SD
Multiparous cows				
DMI (kg)	33.0	4.26	30.0	5.47
BW (kg)	794.2	74.34	822.3	73.57
Actor Behaviors during prepartum period	15.9	7.03	17.0	6.26
No. of different bins visited per day	11.0	0.70	10.9	0.80
No. of bins visited per meal	4.3	1.08	4.4	1.02
Primiparous cows				
DMI (kg)	28.8	2.75	27.2	3.54
Time spent feeding (min)	210.6	25.45	201.5	31.97
No. of bins visited per meal	4.8	0.87	5.2	0.76

^1^ Cows that did not develop ketosis, metritis, and/or mastitis during the postpartum period. ^2^ Cows that became sick with at least one disease (ketosis, metritis, or mastitis) during the postpartum period. ^3^ Dataset that was employed for developing the model. ^4^ Dataset that was employed for testing the model. BW: body weight. DMI: feed intake.

**Table 2 animals-10-00928-t002:** Results from the multivariable models, separately for multiparous and primiparous cows. Estimates of slope, SE, *p*-value, odds ratio (OR), and 95% confidence interval (CI) are reported. Odds ratios for kg DMI, BW, actor behavior, bins visited/day, bins visited/meal, and time spent feeding are calculated based on units of 1 kg DMI, 25 kg BW, 6 actor behaviors, 1 bin visited/day, 1 bin visited/meal, and 15 min time spent feeding, respectively.

Variable Name	Slope	SE	*P*-Value	OR (95% CI)
Multiparous cows				
Kg of feed intake	−0.143	0.066	0.04	^1^
BW prepartum	0.011	0.003	<0.01	1.31 (1.10–1.55)
Actor behavior	0.151	0.043	<0.01	2.47 (1.48–4.13)
Different bins visited per day	−3.354	0.930	<0.01	^1^
Bins visited per meal	0.614	0.273	0.03	^1^
Kg of feed intake * Different bins visited per day	0.279	0.092	<0.01	0.72 (0.61–0.85) ^2^
Bins visited per meal * Different bins visited per day	−2.020	0.543	<0.01	7.83(2.74–22.39) ^3^
Primiparous cows				
Time spent feeding in min	−0.211	0.098	0.03	^1^
Kg of feed intake	−1.641	0.738	0.03	^1^
Bins visited per meal	0.861	0.379	0.02	2.37 (1.13–4.97)
Time spent feeding in min * Kg of feed intake	0.008	0.004	0.04	0.77 (0.60–0.99) ^4^

^1^ Presence of interactions precludes computation of main-effects odds ratios. ^2^ Odds ratio was calculated based upon a 1-unit (kg) increase in feed intake when the number of different bins visited per day was 10.5 (25th percentile of different bins visited per day). ^3^ Odds ratio was calculated for 1 additional bin visited per meal when number of different bins visited per day was 10.5 (25th percentile of different bins visited per day). ^4^ Odds ratio calculated for an increase in every 15 min of time spent feeding when feed intake was 26.4 Kg (25th percentile of feed intake).

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
