# Peer review of "Predicting Disease in Transition Dairy Cattle Based on Behaviors Measured Before Calving"

_animals, 2020, doi:10.3390/ani10060928_

Round 1
Reviewer 1 Report
Firstly, I want to thank you for giving me the opportunity to read and review this manuscript; nicely described study that addresses a stimulating topic for the dairy industry.
I think that the aim and results of this article are interesting.
I found only some notes before its acceptance.
I recommend minor revision of the manuscript.
Language in general: The manuscript is written very well.
Statistical analysis
I have no specific comments for this section, since I am not a statistician, but my impression is that the chosen methods undoubtedly are appropriate for this type of data.
Table 1
Please, check the footnotes of Table 1. In the table caption healthy1 and sick2 animals have inverted footnotes.
Figure 1
I think you should specify in the figure caption that data were from primiparous cows only.
Results
Line 235: please, correct “fore” with “for”.
Discussion
The interpretation and discussion of results seem fully adequate to the aim of the study.
Conclusion
Line 365: should it be “less time feeding during the prepartum period” in place of “more time…”, as shown in Figure 1 and at lines 226-236, where you stated “… increased time spent feeding was protective against becoming sick”?
References
References must be numbered in order of appearance in the text and listed individually at the end of the manuscript.
Author Response
Au: Many thanks to both reviewers for their constructive comments and suggestions. We found these helpful and believe the resulting changes have improved our manuscript. The changes are highlighted in the manuscript for your convenience.
Table 1
Please, check the footnotes of Table 1. In the table caption healthy1 and sick2 animals have inverted footnotes.
Au: Thank you for noticing this. We have now corrected it as you suggested.
Figure 1
I think you should specify in the figure caption that data were from primiparous cows only.
Au: We have now included the term “primiparous” in the figure caption. The new text reads as follows:
Figure 1. Panel A shows the predicted probabilities for postpartum disease in primiparous animals on the Y axis and feed intake on the X axis. The solid blue line represents a primiparous cow that spent 185 minutes feeding (i.e., the 25th percentile), and the red line represents a primiparous cow that spent 222 minutes of feeding (i.e., the 75th percentile). Panel B shows the predicted probabilities for postpartum disease in primiparous animals on the Y axis and time spent feeding on the X axis. The solid blue line represents a primiparous cow that consumed 26.6 kg of feed (i.e., the 25th percentile), and the red line represents a primiparous cow that consumed 30.3 kg (i.e. the 75th percentile). For both panels the variable bins per meal was held constant at the mean value of 5.02.
Results
Line 235: please, correct “fore” with “for”.
Au: Thank you. We have now corrected it. The new line number is 240.
Conclusion
Line 365: should it be “less time feeding during the prepartum period” in place of “more time…”, as shown in Figure 1 and at lines 226-236, where you stated “… increased time spent feeding was protective against becoming sick”?
Au: Thank you for this comment. We have now changed that sentence. The new sentence in line number 373-378 is as follows:
For primiparous cows, key predictors of disease were more bins visited per meal, and an interaction between feed intake and time spent feeding during the prepartum period, where increased time spent feeding was protective against becoming sick when feed intake was low and the probability of becoming sick increased as time spent feeding increased when feed intake was high.
References
References must be numbered in order of appearance in the text and listed individually at the end of the manuscript.
Au: Thank you for picking up on this. It has now been corrected throughout.
Reviewer 2 Report
This is an excellent Ms. It was a highlight to read it. Anyway some suggestion are in the attached file. Please refer to them. Congratulation.

Author Response
Au: Many thanks to both reviewers for their constructive comments and suggestions. We found these helpful and believe the resulting changes have improved our manuscript. The changes are highlighted in the manuscript for your convenience.
Reviewer 2
14 Since You make a prediction, I would like to know also the positive and negative predictive value, because in a clinical setting these figures are more important than sensitivity and specifity. This should be already stated in the simple summary.
Au: Good point. We have now changed our sentence as below. The new line numbers are 13-15.
Our models had high sensitivity (73-71%), specificity (80-84%), positive predictive value (73-77%) and negative predictive value (80-80%) for both cows that had previously calved and for those calving for the first-time.
23 70:30 for what?
Au: thank you for this comment. The 70:30 ratios refer to percentages of data; we used 70% of our data to develop the model and the remaining 30% to test the predictive ability of the model. Lines 23-26 are now modified to read:
The data were split using a stratified random method: we used 70% of our data (hereafter referred to as the ‘training’ dataset) to develop the model and the remaining 30% of data (i.e., the ‘test’ dataset) to assess the model’s predictive ability. Separate models were developed for primiparous and multiparous animals.
82 Either explain Y2K or transfer it into the parenthesis.
Au: Thank you for this comment. We have now moved this specification into the parentheses as seen in line 83.
The dimensions of the stalls were as follows: 2.6 x 1.2 m; neck rail 1.2 m above stall surface; separated by freestall partitions (Y2K partitions; Artex, Langley, BC, Canada).
105 This sentence can be omitted because treatment is not in the focus of this Ms as You mentioned above, but it is not disturbing when it remains in the Ms.
Au: We see your point. After some consideration, we feel that it is probably best to keep this sentence in the manuscript. We need to specify that the same animals were used in the meloxicam study (and thus, some readers may wonder whether we accounted for the meloxicam treatment.)
124 Please describe in more detail the split by 30 : 70.
Au: Lines 126-128 are now revised to read:
Data were split into datasets containing 70% and 30% of observations, according to a stratified random sampling procedure [36]. The stratification was conducted as follows: cows were first grouped as 1) healthy…
126 Please describe the intention of dividing in a training and test data set more exactly.
Au: This is a good point. In an attempt to better clarify this point, we modified the sentence at the end the same paragraph (lines 137 – 139) as follows:
For each parity category, the 70% subset was employed for model development (“training”) and the 30% subset was used to test the model’s predictive ability.
143 Explain in more detail agonistic interaction
Au: Thank you for this comment. We have now changed this section to the following (now lines 144-148):
These variables were feed intake (DMI), time spent eating, number of meals per d, bins visited per d, lameness, BW, and number of agonistic interactions during the prepartum period. Agonistic interactions were defined as any aggressive interaction between individuals, characterized by “actor behaviors” (in which the subject replaces another cow at the feedbunk) and “reactor behaviors” (in which the subject is displaced and replaced by another cow at the feedbunk).
166 This sentence can go to the results
Au: Thank you for suggesting this. We moved this sentence to the results section.
233 By whom? Other cows or by management?
Au: By other cows. We have amended this in the text (Line 238.)
359 Other systems could be mentioned like rumination halter.
Au: Thank you for this comment. We amended this sentence to read:
Automation of data collection of the variables considered here could be achieved by installing automatic feed bins, but these may not be practical for many farmers. More economical alternatives could include the use of sensors to monitor rumination, or water intake (as fewer drinking spaces are required per cow).